# Added diagnostic values of three-dimensional high-resolution proton density-weighted magnetic resonance imaging for unruptured intracranial aneurysms in the circle-of-Willis: Comparison with time-of-flight magnetic resonance angiography

**Younghee Yim**[1], **Seung Chai Jung**[2]*, **Jung Youn Kim**[3], **Seon-Ok Kim**[4], **Byung Jun Kim**[5], **Deok Hee Lee**[2], **Wonhyoung Park**[6], **Jung Cheol Park**[6], **Jae Sung Ahn**[6]

1 Department of Radiology, Chung-Ang University Hospital, Seoul, Korea, 2 Department of Radiology and Research Institute of Radiology, University of Ulsan College of Medicine, Asan Medical Center, Seoul, Korea, 3 Department of Radiology, Kangbuk Samsung Hospital, Seoul, Korea, 4 Department of Clinical Epidemiology and Biostatistics, University of Ulsan College of Medicine, Asan Medical Center, Seoul, Korea, 5 Department of Radiology, University of Korea College of Medicine, Seoul, Korea, 6 Department of Neurosurgery, University of Ulsan College of Medicine, Asan Medical Center, Seoul, Korea

* dynamics79@gmail.com

## Abstract

### Background

Advanced imaging methods can enhance the identification of aneurysms of the infundibula, which can reduce unnecessary follow-ups or further work-up, fear, and anxiety in patients.

### Purpose

This study aimed to evaluate the added diagnostic value of three-dimensional proton density-weighted vessel wall magnetic resonance imaging (3D-PD MRI) in identifying aneurysms from index lesions refer to vascular bulging lesions without vessels arising from the apex, observed using volume-rendered TOF-MRA in the circle-of-Willis compared with time-of-flight magnetic resonance angiography (TOF-MRA).

### Study type

Retrospective.

### Population

A total of 299 patients who underwent 3D-PD MRI, digital subtraction angiography (DSA), and TOF-MRA between January 2012 and December 2016 were retrospectively enrolled in this study.

**Data Availability Statement:** All relevant data are within the manuscript and its Supporting information files.

**Funding:** This study was supported by the National Research Foundation of Korea (NRF) grant funded by the Korea government (2019R1A2C1089939).

**Competing interests:** The authors have declared that no competing interests exist.

### Field strength/Sequence

3 Tesla, 3D-PD MRI.

### Assessment

Three neuroradiologists independently evaluated TOF-MRA and 3D-PD MRI combined with TOF-MRA for the determination of intracranial aneurysms in patients with index lesions within the circle of Willis. Final diagnoses were made by another neuroradiologist and neurointerventionist in consensus using DSA as the reference standard. The diagnostic performance and proportions of undetermined lesions on TOF-MRA and 3D-PD MRI with TOF-MRA were assessed based on the final diagnoses.

### Statistical tests

The sensitivity, specificity, positive predictive value, negative predictive value, and accuracy for the diagnosis of unruptured intracranial aneurysms were calculated for each imaging modality.

### Results

Of 452 lesions identified on volume-rendered TOF-MRA images, 173 (38%) aneurysms and 276 (61%) infundibula were finally diagnosed on DSA. 3D-PD MRI with TOF-MRA showed superior diagnostic performance (p = .001; accuracy, 85.5% versus 95.4%), superior area under the receiver operating characteristic curve over TOF-MRA (p = .001; 0.837 versus 0.947), and a lower proportion of undetermined lesions than TOF-MRA (p = .001; 25.1% versus 2.3%).

### Data conclusion

For unruptured intracranial aneurysms in the circle of Willis, 3D-PD MRI can complement TOF-MRA to improve diagnostic performance and lower the proportion of undetermined lesions.

### Introduction

Intracranial aneurysms occur in approximately 3%–5% of the general population [1,2]. With the increase in the demand for health checkups and accessibility to advanced imaging techniques, the detection of incidental aneurysms is increasing [3,4]. However, small infundibular dilatations are sometimes confused with true aneurysms [3,5]. Infundibula are detected on angiography or autopsy with occurrence rates ranging from 7% to 25%, and are generally considered to be benign lesions that do not require treatment or follow-up [4,6]. However, as untreated intracranial aneurysms have a risk of rupture, the fear and anxiety caused by the knowledge of having an aneurysm can affect the quality of life, and this therefore underlines the need for accurate differentiation of infundibula from aneurysms [7].

Catheterized digital subtraction angiography (DSA) is the gold standard for the definitive diagnosis of intracranial aneurysms; however, the procedure carries a risk of developing complications [8,9]. Although time-of-flight magnetic resonance angiography (TOF-MRA) is currently widely used for the screening of unruptured intracranial aneurysms owing to its noninvasiveness, the definitive diagnosis of aneurysms or infundibula can sometimes be

challenging, in particular in small lesions [4]. The definitive diagnosis of small lesions suspected of being aneurysms in the circle of Willis may sometimes be challenging because of the high incidence of infundibula in the posterior communicating artery and anterior choroidal artery origin, the high incidence of anatomic variation such as fenestration in the anterior communicating arteries and the limited resolution on TOF-MRA [10–14]. High-resolution magnetic resonance imaging (HR-MRI) for vessel walls with a 3-T scanner has become a useful imaging modality for characterizing and diagnosing intracranial arterial steno-occlusive lesions, providing detailed information on vessel wall morphology [15,16] and a high correlation with DSA [17–19]. Proton density-weighted vessel wall MRI (3D-PD MRI) can depict even tiny vascular structures with high signal-to-noise ratios among commonly used vessel wall MRI sequences [20]. There was one report that demonstrated the performance of proton density images for identifying aneurysms of the infundibula. However, Kim et al. discussed that the diagnostic performance of 3D-PD MRI was found to be better than TOF-MRA only in the lesions of the posterior communicating arteries within a small population [21]. Yoon et al. found that PD-MRI can disclose the detailed location (intradural, extradural, or transdural) of paraclinoid internal carotid artery aneurysms with reference to distal dural rings without identifying aneurysms of the infundibula [22]. We hypothesized that 3D-PD MRI would be useful in accurately identifying true unruptured aneurysms from the index lesion. Therefore, this study aimed to evaluate the added diagnostic value of 3D-PD MRI in identifying aneurysms from index lesions (vascular bulging lesions without vessels arising from the apex) within the circle of Willis and thus compare between 3D-PD MRI combined with TOF-MRA and TOF-MRA alone using DSA as a reference standard.

## Materials and methods

### Study population

This retrospective study was approved by Asan Medical Center institutional review board, which waived the requirement for informed consent. A total of 299 patients who underwent 3D-PD MRI, DSA, and TOF-MRA between January 2012 and December 2016 were retrospectively enrolled. Patients who 1) underwent 3D-PD MRI, TOF-MRA, and DSA and 2) had index lesions (vascular bulging without vessels arising from the apex on volume-rendered TOF-MRA) and thus suspected of having intracranial aneurysms within the circle of Willis (target arteries: anterior communicating artery, bilateral A1 segment of anterior cerebral arteries, origin of bilateral ophthalmic arteries, origin of bilateral posterior communicating arteries, origin of bilateral anterior choroidal arteries, origin of bilateral superior cerebellar arteries, and top of basilar artery) were included. By contrast, patients 1) with no lesion within the target arteries on TOF-MRA (704 patients), 2) whose 3D-PD MRI scan parameters did not include the target arteries (163 patients), 3) whose 3D-PD MRI had severe artifact due to previous coil embolization or clipping that hindered evaluation of the target arteries (29 patients), and 4) whose imaging studies did not contain TOF-MRA source images (15 patients) were excluded. Finally, 452 index lesions within the circle of Willis in 299 patients were included in this study. Typical paraclinoid internal carotid artery (ICA) aneurysms with inferomedial directions were considered nontarget lesions and were not included, because it is easy to differentiate between such aneurysms and infundibula on TOF-MRA. Index lesions with an ophthalmic artery origin are lesions that appeared to be in contact with the ophthalmic arteries on volume-rendered TOF-MRA images. The mean time interval between all modalities was 29.8 days (ranges: <1 month, 203; 1–6 months, 96). Demographic and clinical data were collected by reviewing the patients' digital medical records (Table 1). This study was reported in accordance with the Standards for Reporting of Diagnostic Accuracy Studies guidelines [23].

**Table 1. Patients' demographics and lesion details.**

| Variables | | |
|---|---|---|
| **Final diagnosis of index lesions** | | |
| Aneurysm | 173 (38.0%) | |
| Infundibulum | 276 (61.0%) | |
| No lesion | 3 (1.0%) | |
| | Index lesions | Aneurysms |
| **Age (y)** * | 49.4 ± 10.2 | 46.8 ± 13.6 |
| **Male: Female** | 38:261 | 17:123 |
| **Size** | | |
| < 3 mm | 302 (66.8%) | 53 (30.6%) |
| ≥ 3mm and ≤ 5 mm | 91 (20.1%) | 64 (37.0%) |
| > 5 mm | 59 (13.1%) | 56 (32.4%) |
| **Location** | | |
| Posterior communicating artery | 239 (52.9%) | 44 (25.4%) |
| Ophthalmic artery | 97 (21.5%) | 77 (44.5%) |
| Anterior choroidal artery | 62 (13.7%) | 15 (8.7%) |
| Anterior communicating artery | 45 (10.0%) | 28 (16.2%) |
| Top of basilar artery | 4 (0.9%) | 4 (2.3%) |
| Superior cerebellar artery | 2 (0.4%) | 2 (1.15%) |
| Internal carotid artery bifurcation | 2 (0.4%) | 2 (1.15%) |
| Anterior cerebral artery, A1 | 1 (0.2%) | 1 (0.6%) |

* Mean ± standard deviation.

† Numbers in parentheses indicate the proportion.

## Image acquisition

TOF-MRA and 3D-PD MRI were simultaneously performed using the same scanner. TOF-MRA and 3D-PD MRI were performed on 3-T MR systems (Ingenia CX with a 32-channel head coil, Philips Healthcare, Netherlands; Skyra with a 64-channel head coil, Siemens, Erlangen, Germany; Achieva with a 6-channel head coil, Philips Healthcare).

The TOF-MRA parameters were as follows: multiple overlapping thin slabs acquisition (MOTSA), 2/5/2 thin slabs (100/200/100 slices); tilted optimized nonsaturating excitation (TONE), on; magnetization transfer contrast (MTC), off; repetition time (TR), 25/22/25 ms; echo time (TE), 3.5/3.71/3.5 ms; flip angle, 20/18/20˚; section thickness, 1/0.64/1 mm; matrix, 880× 880 × 100/575 × 433 × 228/880 × 880 × 100 mm; field of view (FOV), 200 × 200 × 50/230 × 173 × 137/200 × 200 × 50 mm; voxel size, 0.23 ×0.23 × 0.5/0.4 × 0.4 × 0.6/0.23 ×0.23 × 0.5 mm; number of excitations (NEX), 1/1/1; acceleration factor, 2/2/2; total acquisition time, 5 min 58 s/5 min 16 s/5 min 58 s. Maximum intensity projection (MIP) and volume rendered images were created for each data set for evaluation of the intracranial aneurysms. Initially, for each of the MIP image sets, the anterior and posterior circulations were interactively rotated around the axial and sagittal axes at 15˚ increments to examine for the presence of harbored aneurysms. Further arbitrary oblique projections were also obtained when necessary, to overcome vascular overlapping.

The 3D-PD MRI acquisitions were made using axial 3D turbo spin-echo sequences and 3D reconstructions with the following parameters: TR, 2000 ms; TE, 35.4/21.0/30.7 ms; flip angle, 90˚/125˚/90˚; matrix, 640 × 640 × 150/640 × 640 × 320/640 × 640 × 100; FOV, 120 × 120 × 30/

$160 \times 160 \times 80/180 \times 180 \times 30$ mm$^3$; voxel size, $0.2 \times 0.2 \times 0.2/0.3 \times 0.3 \times 0.3/0.3 \times 0.3 \times 0.3$ mm$^3$; number of excitations, 1/2/1; total acquisition time, 12 min 36 s/5 min 10 s/5 min 38 s.

DSA was performed on a biplane system (Artis Zee, Siemens) with a diagnostic angiography catheter being introduced into the ascending aorta via the transfemoral route and navigated into the appropriate carotid or vertebral artery as decided by the neurointerventionists. With the injection of contrast media, the intracranial arteries were displayed in at least two projections (i.e., anteroposterior and lateral) and/or as 3D rotational angiography. The contrast injection rate for angiography with a catheter in ICA was 4–5 ml/s (Mark V ProVis; Medral, Warrendale, PA, USA), with a total volume of 7–8 ml and in vertebral artery was 4–5 ml/s with a total volume of 8–10 ml. DSA was performed with a 320-mm FOV and a 1024 or 2018 matrix, yielding a pixel size of $0.315 \times 0.315$ mm$^2$ or $0.159 \times 0.159$ mm$^2$, respectively.

## Image analysis

One neuroradiologist (X.X.X. with 7 years of experience) identified index lesions (vascular bulging lesions with unclear or no demonstration of vessels arising from the apex on volume-rendered TOF-MRA images) and provided observers with anatomic location information on the index lesion. Then three neuroradiologists (X.X.X. with 2 years of experience, X.X.X. with 2 years of experience, and X.X.X. with 14 years of experience) independently classified the index lesions as definite aneurysm, infundibulum, or undetermined lesions on TOF-MRA alone (volume rendering and source images) and on 3D-PD MRI combined with TOF-MRA. They reviewed the images with 2-week intervals on TOF-MRA and 3D-PD MRI combined with TOF-MRA to avoid recall bias. The review order of the imaging methods was decided by the observer. One neuroradiologist (X.X.X. with 7 years of experience) and one neurointerventionist (X.X.X. with 5 years of experience) made the final diagnoses of the index lesions as definite aneurysm, infundibulum, or no lesion, on either DSA (n = 452) or with 3D rotational angiography (n = 324), but based on consensus. In order to classify the index lesions using TOF-MRA and 3D-PD MRI, the following criteria were applied: Aneurysm was defined as a saccular protrusion from the side wall of the cerebral arteries [24]; infundibulum was defined as a funnel-shaped vascular enlargement at the origin of the cerebral arteries [4]; undetermined lesion was defined as insufficient findings to decide on a definite diagnosis. The diameters of index lesions were manually measured on DSA using electronic calipers (X.X.X. with 2 years of experience) [25]. The image quality for TOF-MRA was assessed using the following visual scoring system: 5, excellent; 4, more than adequate for diagnosis; 3, adequate for diagnosis; 2, less than adequate for diagnosis; and 1, nondiagnostic [26]. The image analysis was performed using a picture archiving and communication system (PACS) workstation.

## Statistical analysis

The summary statistics were presented as numbers and percentages for categorical variables and means with standard deviations for continuous variables. The sensitivity, specificity, positive predictive value, negative predictive value, and accuracy for the diagnosis of unruptured intracranial aneurysms were calculated for each imaging modality (TOF-MRA and 3D-PD MRI) and the size of the lesions was determined on a per-lesion basis. The diagnostic performance was analyzed using pooled data from three observers. The infundibula, no lesion, and undetermined lesions were not classified as aneurysm (negative result) and thus were not included in the statistical analyses. DSA was used as the reference standard to assess diagnostic performance. Diagnostic performance among imaging modalities was compared by generalized estimating equations (GEE) that accounted for the clustering of the same patient. A non-parametric method for clustered data (multiple lesions and multiple readers) proposed by

Obuchowski was used to compare areas under the receiver operating characteristic (ROC) curve. The proportions of undetermined lesions were calculated and compared according to the modalities using DSA as a reference standard. 3D-PD MRI combined with TOF-MRA was compared with TOF-MRA alone in 299 patients in terms of diagnostic performance and proportions of undetermined lesions. The inter-reader agreement for the diagnosis of index lesions was evaluated using the Cohen's kappa statistic. A κ-value <0 indicated no agreement, whereas 0–0.20, 0.21–0.40, 0.41–0.60, 0.61–0.80, and 0.81–1 indicated slight, fair, moderate, substantial, and almost perfect agreement, respectively [27]. All statistical analyses were performed using SAS version 9.4 (SAS Institute, Cary, NC, USA) and R version 3.4.2. Two-sided p-values less than .05 were considered significant.

## Results

Of the 452 lesions identified on volume-rendered TOF-MRA images, 173 (38.0%) were diagnosed on DSA as aneurysms, 276 (61.0%) as infundibula, and 3 (1.0%) as non-lesions. The median lesion size and range were 2.5 mm and 1.0 to 20 mm, respectively (Table 1). Representative cases are illustrated in Figs 1–3.

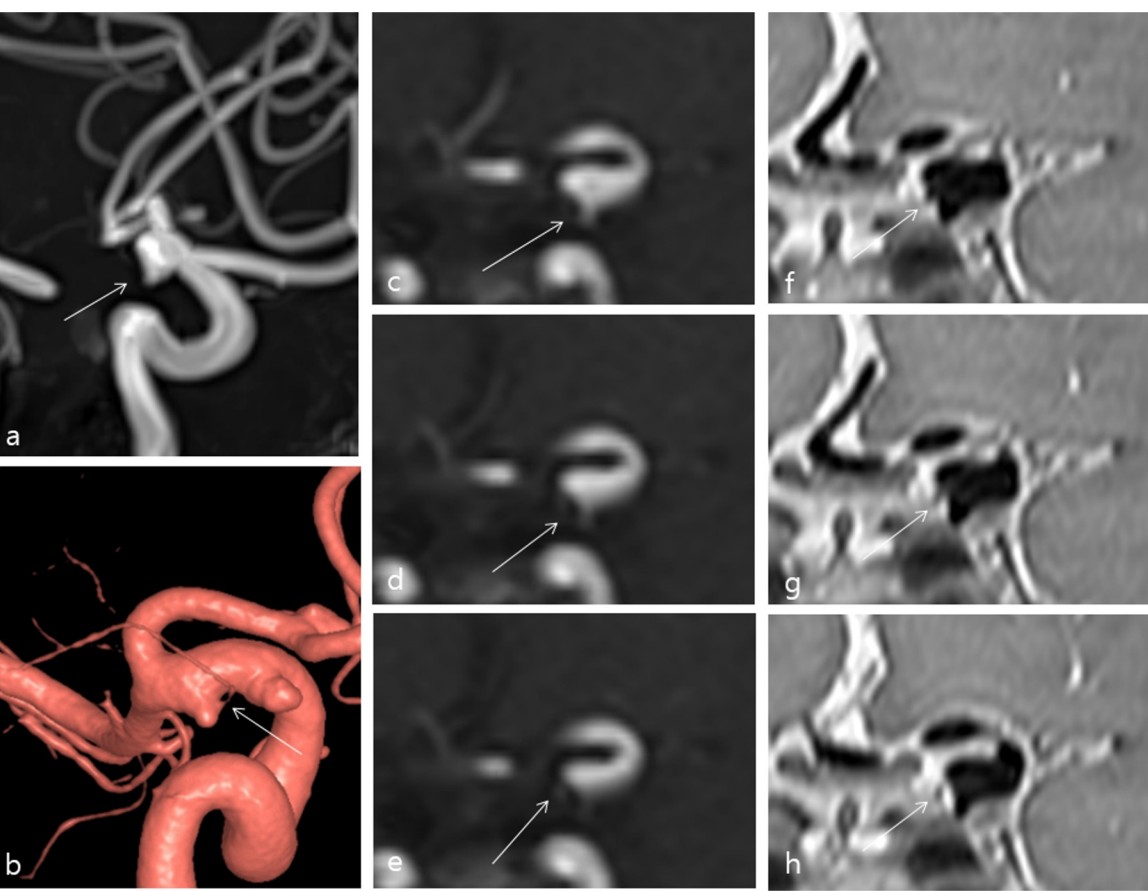

**Fig 1. A 60-year-old female who was referred from an outside hospital for incidentally found aneurysms.** TOF-MRA (a) and TOF-MRA source images (c-e) showed an index lesion around the left anterior choroidal artery origin; however, the relationship between the aneurysm and anterior choroidal artery was unclear on coronal multi-planar reconstruction (MPR) images (arrows). 3D-PD MRI clearly depicted the aneurysm and anterior choroidal artery origin on coronal MPR images (arrows) (f-h). 3D rotational angiography (b) also showed a clear relationship between the aneurysm and the anterior choroidal artery origin (arrow).

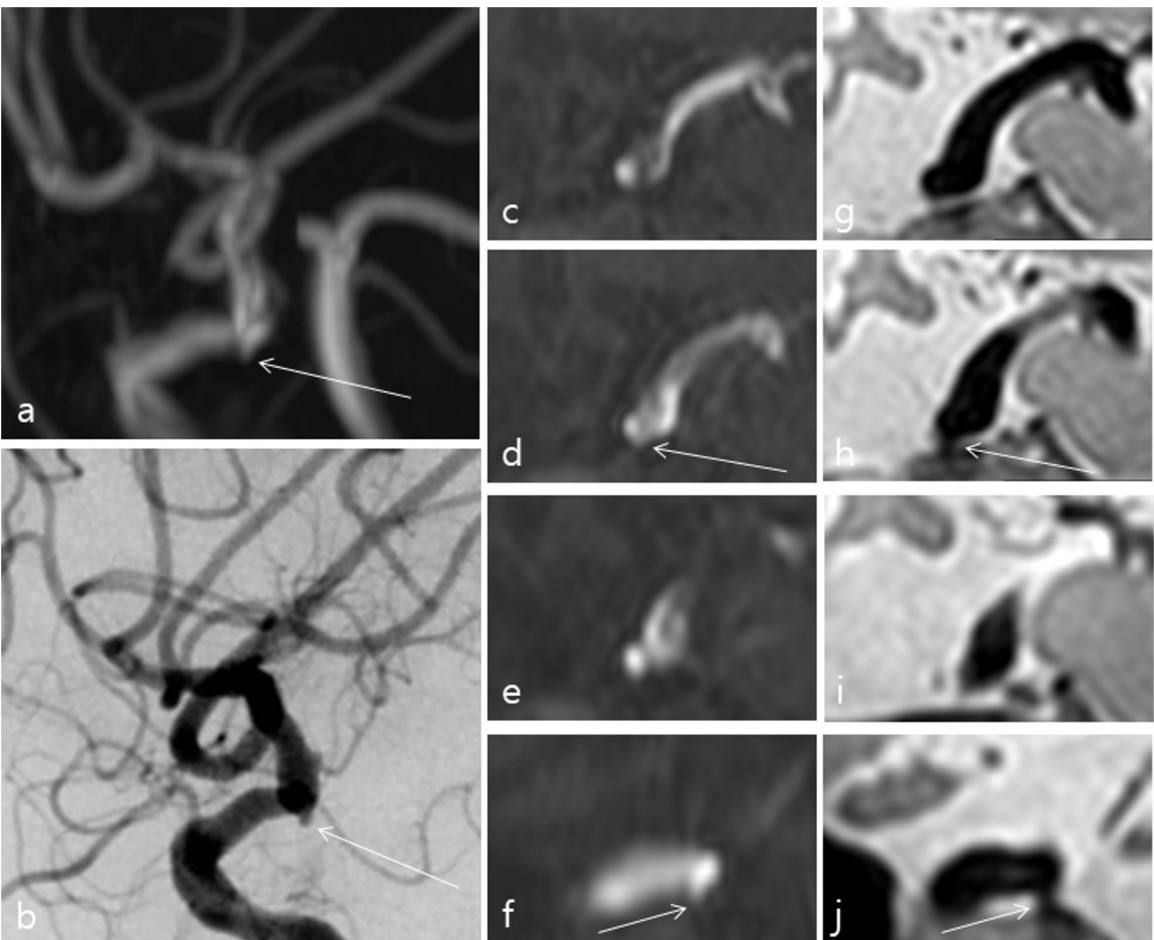

**Fig 2. A 66-year-old female who underwent 3D-PD MRI for evaluation of a right middle cerebral artery stenosis detected during work up for left side weakness.** TOF-MRA (a) showed an index lesion around the left distal ICA. However, it was difficult to diagnose the aneurysm due to its tiny size and nearby bone structures (arrows) (c–f) even on TOF-MRA source images with oblique coronal multi-planar reconstruction (MPR) (d-f). 3D-PD MRI depicted a definite aneurysm on oblique coronal MPR images (arrows) (g-j). DSA also showed a definite aneurysm (arrow) (b).

### Diagnostic performance of TOF-MRA alone and 3D-PD MRI combined with TOF-MRA

The sensitivity, specificity, positive predictive value, and negative predictive values of TOF-MRA alone were 77.6%, 90.4%, 83.4%, and 86.7%, respectively. The sensitivity, specificity, positive predictive value, and negative predictive values of 3D-PD MRI combined with TOF-MRA were 91.5%, 97.8%, 96.3%, and 94.9%, respectively (Table 2).

### Comparison of diagnostic performance

3D-PD MRI combined with TOF-MRA showed higher diagnostic performance and AUC than TOF-MRA alone (p = .001) (Tables 2 and 3, and Fig 4). 3D-PD MRI combined with TOF-MRA showed a larger AUC compared to that of TOF-MRA alone in the ophthalmic artery origin, posterior communicating artery, anterior communicating artery, and anterior choroidal artery origin locations (p = .033 to .001).

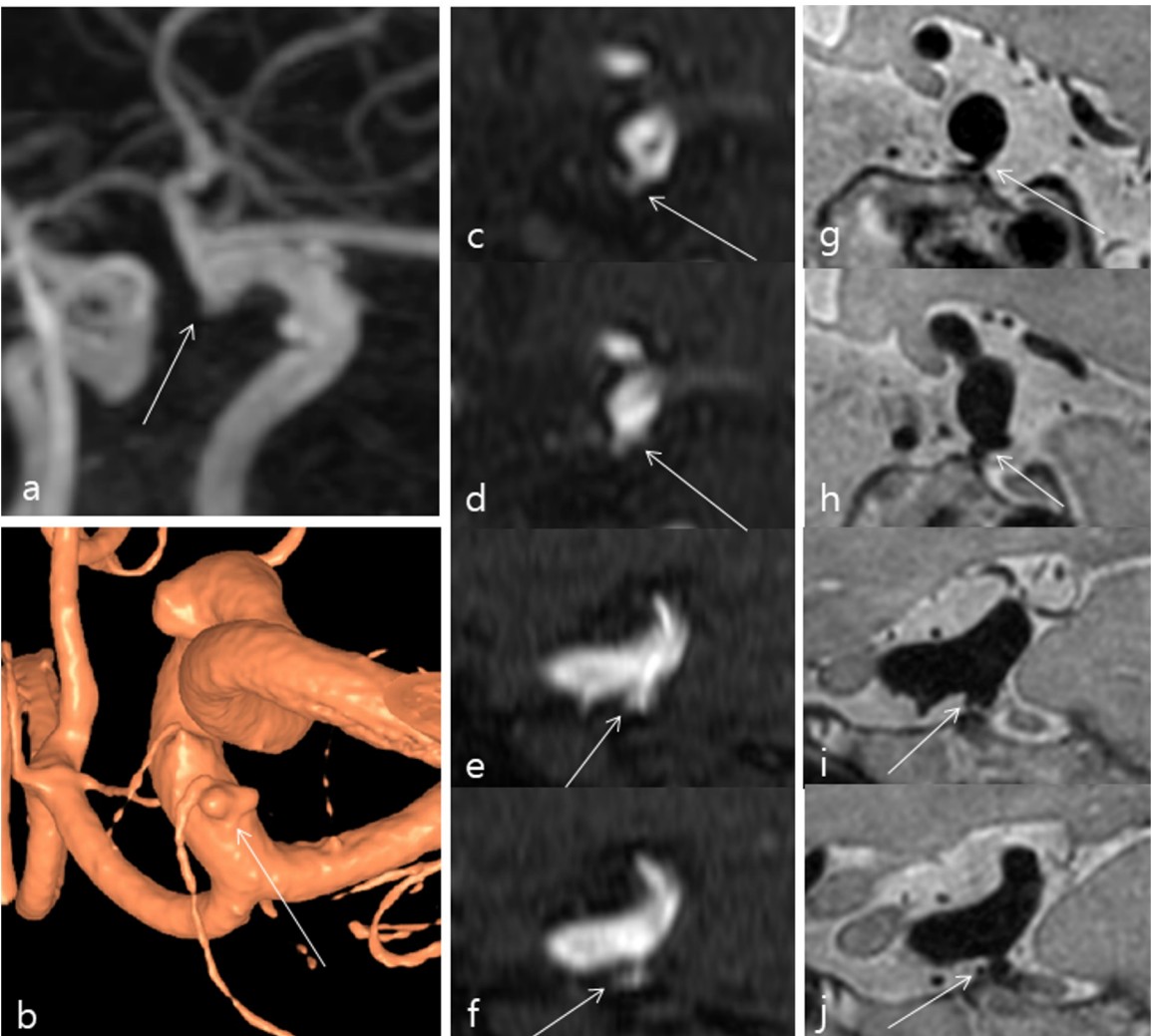

**Fig 3. A 57-year-old female underwent 3D-PD MRI for evaluation of a left paraclinoid internal carotid artery aneurysm.** TOF-MRA (a) showed an index lesion around anterior choroidal artery origin of left distal ICA. However, it was difficult to diagnose the aneurysm (arrows) even on TOF-MRA source images with sagittal multi-planar reconstruction (MPR) images (c-f). The relationship between anterior choroidal artery and the index lesion was not seen clearly on sagittal MPR images of TOF-MRA (e, f). 3D-PD MRI depicted a definite aneurysm with bi-lobed shape (arrows) (g, h) and clear relationship between anterior choroidal artery and the index lesion on sagittal MPR images (i, j). 3D rotational angiography also showed a definite aneurysm with bi-lobed shape (arrow) (b).

### Proportions of undetermined lesions

The proportions of undetermined lesions on 3D-PD MRI with TOF-MRA were lower than those on TOF-MRA alone (p = .001) (Table 3).

Interobserver agreement for identifying aneurysms from index lesions among the three observers was 0.793 (TOF-MRA, 0.679; 3D-PD MRI combined with TOF-MRA, 0.873). The mean scores for the image quality were 4.1 for TOF-MRA.

## Discussion

Compared with TOF-MRA, 3D-PD MRI had added diagnostic values in identifying true aneurysms from index lesions in the circle of Willis and lower proportions of undetermined lesions.

**Table 2. Diagnostic performances of TOF-MRA and 3D-PD MRI combined with TOF-MRA.**

| TOF-MRA | | | | | | | | | |
|---|---|---|---|---|---|---|---|---|---|
| | TP | TN | FP | FN | Sensitivity | Specificity | PPV | NPV | Accuracy |
| | 403 | 757 | 80 | 116 | 77.6 (.001) | 90.4 (.001) | 83.4 (.001) | 86.7 (.001) | 85.5 (.001) |
| **3D-PD MRI with TOF-MRA** | | | | | | | | | |
| | TP | TN | FP | FN | Sensitivity | Specificity | PPV | NPV | Accuracy |
| | 475 | 819 | 18 | 44 | 91.5 | 97.8 | 96.3 | 94.9 | 95.4 |

* TOF-MRA = time-of-flight magnetic resonance angiography, 3D-PD MRI = three-dimensional high-resolution proton density-weighted magnetic resonance imaging, TP = true-positive, TN = true-negative, FP = false-positive, FN = false-negative, PPV = positive predictive value, NPV = negative predictive value, PPV = positive predictive value, NPV = negative predictive value.

† Data are percentages, and parentheses indicate p-values compared with 3D-PD MRI with TOF-MRA.

Therefore, 3D-PD MRI can play a role as a supplementary imaging modality for the diagnosis of aneurysms.

TOF-MRA showed an excellent diagnostic performance for diagnosing intracranial aneurysms [28]. However, the accurate diagnosis of intracranial aneurysms is sometimes quite challenging, especially in lesions with a small size and a high possibility of normal variation and/or infundibula [3,4]. Common locations of infundibular dilatation in intracranial arteries are the posterior communicating artery and the anterior choroidal artery; hence, it is often necessary to differentiate infundibular dilatation from small aneurysms [10,11].

HR-MRI for vessel walls is generally used for the evaluation of arterial walls in steno-occlusive lesions or aneurysms [15,16]. This study of 299 patients demonstrated the usefulness of 3D-PD MRI compared to that of TOF-MRA in identifying infundibular aneurysms in entire arterial segments of the Circle of Willis, except the paraclinoid internal carotid artery aneurysms. However, Kim et al. presented a pilot study of 82 patients investigating only the lesions of the posterior communicating arteries [21]. Therefore, 3D-PD MRI can be applied widely [21]. Therefore, 3D-PD MRI can be applied widely. In clinical practice, there have been few imaging options for the diagnosis of intracranial aneurysms beyond TOF-MRA and CTA. Thus, DSA is still mobilized for the definitive determination, despite its invasiveness with neurologic complications, radiation exposure, and iodinated contrast media [29]. Proton density imaging is a commonly used vessel wall MRI technique with a high signal-to-noise ratio that can present tiny vascular structures compared with other vessel wall MRI sequences such as

**Table 3. Comparison of diagnostic performances and proportions of undetermined lesions.**

| | AUC | P value |
|---|---|---|
| TOF-MRA | 0.837 † (0.817–0.857) | **.001** |
| 3D-PD MRI with TOF-MRA | 0.947 † (0.934–0.958) | |
| | Proportion | P value |
| TOF-MRA | 25.1% †† (341/1356) | **.001** |
| 3D-PD MRI with TOF-MRA | 2.3% †† (31/1356) | |

* TOF-MRA = time-of-flight magnetic resonance angiography, 3D-PD MRI = three-dimensional high-resolution proton density-weighted magnetic resonance imaging, AUC = area under receiver operating characteristic curve.

† Data in parentheses indicate 95% confidence intervals.

†† Data in parentheses indicate numbers of lesions.

††† P-values in were calculated from comparisons with 3D-PD MRI with TOF-MRA.

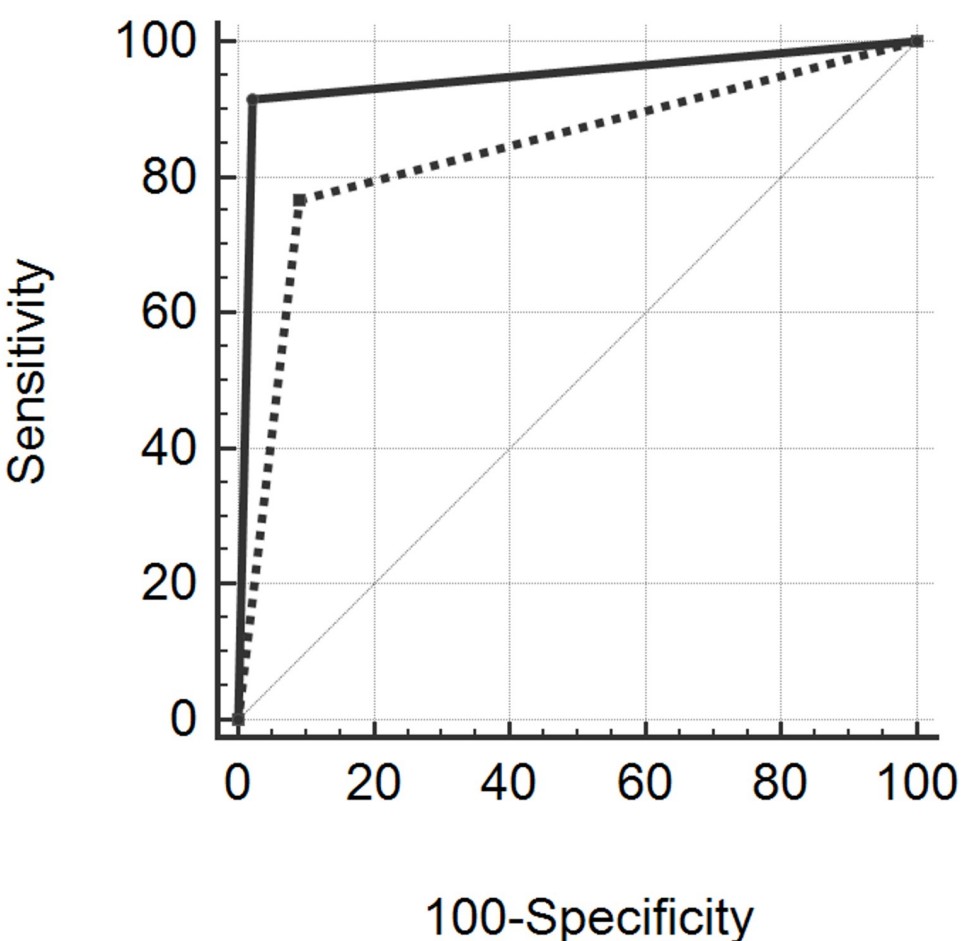

**Fig 4. 3D-PD MRI combined with TOF-MRA (AUC = 0.837) showed higher diagnostic performance and AUC than TOF-MRA alone (AUC = 0.947) (p = .001).**

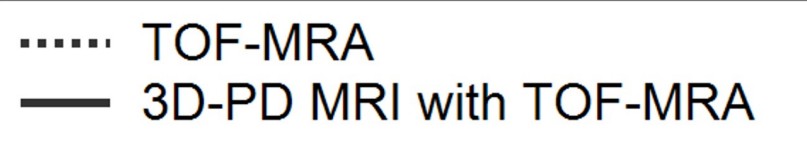

T1- or T2-weighted imaging [20]. Therefore, 3D-PD MRI can depict small perforators or the relation of saccular vessel irregularity to branches such as posterior communicating or anterior choroidal artery origin better. This study thereby introduces 3D-PD MRI as another diagnostic option for intracranial aneurysms.

　　Recently introduced compressed sensing techniques may contribute to the enhancement of spatial resolution with preserved scan times and the reduction of scan time with preserved image quality in both TOF-MRA [30,31] and vessel wall MRI [32]. TOF-MRA with a higher resolution involving a reasonable scan time may enhance the identification of aneurysms of the infundibula even though a smaller spatial resolution can have a demerit of lower signal-to-noise ratio [33]. However, recent research on TOF-MRA has focused on the reduction of scan

times, and not on the enhancement of spatial resolution [30,31]. Moreover, even 7T system shows a resolution of 0.23–0.31 mm$^3$ [4,34]. Therefore, further studies are necessary.

This study had some limitations. First, this study was retrospective in nature and included patients from a single tertiary referral institute. The study cannot present the actual diagnostic performance of 3D-PD MRI as the index lesions evaluated in this study were those previously detected from volume-rendered TOF-MRA images. In addition, the images were not acquired using the same machines or protocols across all patients. The retrospective collection may have contributed to the severely unbalanced proportions of men and women and higher incidence of aneurysms in the general population. Second, the 3D-PD MRI may have originally been performed for the evaluation of aneurysms in other locations (paraclinoid ICA aneurysms and beyond the circle of Willis) and for steno-occlusive lesions; it was not performed for the purposes of this study. Third, this study included incidentally found unruptured intracranial aneurysms within the circle of Willis, and 66.8% of these were less than 3 mm in size. The resulting selection bias may limit the generalizability of our findings on 3D-PD MRI. Fourth, only 28.3% of DSA evaluations were performed with 3D rotational angiography to make the final decision, even though two experienced observers thoroughly reviewed the DSA to diagnose aneurysms. Fifth, the time intervals between 3D-PD MRI and DSA were variable, being up to 6 months in some cases. Hence, it remains unclear whether the changes in size and morphology did not affect the diagnostic performances. Finally, TOF-MRA was acquired with anisotropic voxels and variable parameters, and was observed to have a slightly larger spatial resolution in the z-axis. The heterogeneity of detailed parameters and differences in spatial resolutions may have affected the results of this study. Therefore, further studies involving equal imaging conditions are warranted.

## Conclusions

The 3D-PD MRI can complement TOF-MRA to improve diagnostic performance and lower the proportion of undetermined lesions when identifying aneurysms from the index lesions in the circle of Willis.

## Supporting information

**S1 Data.**
(XLSX)

## Author Contributions

**Conceptualization:** Seung Chai Jung, Jung Youn Kim, Deok Hee Lee.

**Data curation:** Younghee Yim, Seung Chai Jung.

**Formal analysis:** Younghee Yim, Seung Chai Jung.

**Funding acquisition:** Seung Chai Jung.

**Investigation:** Younghee Yim, Seung Chai Jung, Jung Youn Kim.

**Methodology:** Seung Chai Jung, Jung Youn Kim, Seon-Ok Kim, Byung Jun Kim.

**Project administration:** Seung Chai Jung.

**Resources:** Seung Chai Jung, Deok Hee Lee, Wonhyoung Park, Jung Cheol Park, Jae Sung Ahn.

**Software:** Seung Chai Jung.

**Supervision:** Seung Chai Jung.

**Validation:** Younghee Yim, Seung Chai Jung, Jung Youn Kim, Byung Jun Kim.

**Visualization:** Seung Chai Jung.

**Writing – original draft:** Younghee Yim, Seung Chai Jung.

**Writing – review & editing:** Younghee Yim, Seung Chai Jung.

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
