## [Decision Letter · Decision Letter 0]

13 Jul 2020

PONE-D-20-13596

Added diagnostic values of three-dimensional high-resolution proton density-weighted magnetic resonance imaging for unruptured intracranial aneurysms in the circle-of-Willis: Comparison with time-of-flight magnetic resonance angiography.

PLOS ONE

Dear Dr. Jung,

Thank you for submitting your manuscript to PLOS ONE. After careful consideration, we feel that it has merit but does not fully meet PLOS ONE’s publication criteria as it currently stands. Therefore, we invite you to submit a revised version of the manuscript that addresses the points raised during the review process.

This manuscript was evaluated with substantial criticism, in particular with regards of omission of important recent references of very similar studies on PD imaging of intracranial aneurysms by Korean groups:

1. Kim S, Chung J, Cha J, et al. Usefulness of high-resolution three-dimensional proton density-weighted turbo spin-echo MRI in distinguishing a junctional dilatation from an intracranial aneurysm of the posterior communicating artery: a pilot study. Journal of NeuroInterventional Surgery 2020;12:315-319

2. Seon Jin Yoon, Na-Young Shin, Jae Whan Lee, Seung Kon Hu and Keun Young Park. Localization and Treatment of Unruptured Paraclinoid Aneurysms: A Proton Density MRI-based Study. J Cerebrovasc Endovasc Neurosurg. 2015;17(3):180-184

There appears to be a serious concern of dual publication of data and it appears unclear whether these are publications from the same group or not. It is not understandable why the authors did not mention these publications from colleagues on the same topic!

So we cannot accept this paper in PLOS ONE if the authors do not make clear that:

1. there is no overlap in analyzed patient cohorts between their study submitted to our journal and the other two studies published in different journal.

2. what the added values and differences in methods, analyzed endpoints/variables and results between their submitted studied and the two previously published studies was!

In addition, please answer all other comments from both authors and modify your manuscript accordingly. However, we cannot guarantee any publication if the major issue of potential dual publication cannot be solved convincingly by your answer.

We look forward to receiving your revised manuscript.

Kind regards,

Stephan Meckel, MD, PhD

Academic Editor

PLOS ONE

Journal Requirements:

2.  Please clarify in your Data availability statement how other researchers can obtain the same dataset. For PLOS ONE data sharing guidelines, please see " ext-link-type="uri" xlink:type="simple">https://journals.plos.org/plosone/s/data-availability"

3. Thank you for including your ethics statement:

"This retrospective study was approved by our institutional review board, which waived the requirement for informed consent. ".

i) Please amend your current ethics statement to include the full name of the ethics committee/institutional review board(s) that approved your specific study.

ii) Once you have amended this statement in the Methods section of the manuscript, please add the same text to the “Ethics Statement” field of the submission form (via “Edit Submission”).

Reviewers' comments:

Reviewer's Responses to Questions

**Comments to the Author**

1. Is the manuscript technically sound, and do the data support the conclusions?

Reviewer #1: Yes

Reviewer #2: Partly

2. Has the statistical analysis been performed appropriately and rigorously? 

Reviewer #1: Yes

Reviewer #2: No

3. Have the authors made all data underlying the findings in their manuscript fully available?

Reviewer #1: Yes

Reviewer #2: No

4. Is the manuscript presented in an intelligible fashion and written in standard English?

Reviewer #1: Yes

Reviewer #2: Yes

5. Review Comments to the Author

Reviewer #1: This is an interesting original paper that quantifies with statistical analysis the improvement in diagnostic performance of the TOF-MRA combined with 3D-PD MRI with respect to the TOF-MRA performed alone for the identification of aneurysms and infundibula from the detected lesions in the circle-of-Willis. DSA is used as reference standard to verify the diagnostic outcomes. The methodology and the statistics are sufficiently detailed, and the study is properly placed in the context. The Authors clearly expose the motivation for this study and its use in the context of lifting the patient from the fear and anxiety due to the knowledge of having an aneurysm, when the probability of that lesion actually being an infundibulum is not negligible. Also, the proposed technique is completely non-invasive and would offer additional diagnostic capabilities by using it to complement other non-invasive techniques such as TOF-MRA, which used alone would have lower performance. This is an important aspect to take into consideration, since the gold standard alternative of the DSA carries important risks, as explained by the authors.

The image acquisition details used for the different diagnostic techniques are extensively provided by the Authors. However, it would be nice if more information could be added about the use of different imaging parameters with the same technique and in between the techniques, specifying if the changes are compatible with the possibility of a comparison between the techniques in terms of diagnostic capabilities. For example, can a possible change in resolution have affected the results?

A few comments for consideration:

1) At line 317 in the Discussion section the Authors mention that the time elapsed between the PD MRI and the DSA is variable and in some cases up to 6 months. Would it be possible to add an additional comment specifying how this variation could have significantly or not affected the results?

I believe it would also be important at this point to refresh the memory of the readers about the fact that the TOF-MRA and 3D-PD MRI have been performed simultaneously, thus the main comparison between TOF-MRA alone and TOF-MRA combined with 3D-PD MRI is not affected by this time issue.

Regarding the comparison with DSA instead, which would be affected by the time issue, my understanding is that if DSA was performed afterwards, and only a worsening of the situation can be considered with time, the calculated performance compared to DSA could have only been better than the shown results in the case where the DSA had been performed without waiting a long time.

2) At line 7, term “index lesions”. I see the Authors give a definition of the term at line 80 in the Introduction section but, using this term in the abstract without a definition seems confusing to me. I would suggest providing an explanation for this from the beginning.

3) Multiple times throughout the manuscript the Authors use the term “differentiating”. An instance can be found at line 81. The word "differentiating" as it is used in the sentences leads me to think that the Authors want to differentiate between the aneurysm and the index lesion, while they are actually identifying which of the index lesion is an aneurysm or an infundibulum. Therefore, I would rather use the word identifying, by saying "identifying the true unruptured aneurysms among the index lesions".

4) Line 14, “Three neuroradiologists”: after reading this in the abstract, it was confusing for me to find more people mentioned in the Materials and Methods section under Image Analysis.

5) Line 16: Is the parenthesis an explanation of what the Authors mean with aneurysm, differently from the infundibula from which instead vessel originates? As it is positioned in the sentence, it seems instead referring to the definition of index lesions.

6) End of the line 18: I think there is an “of” that would need to be deleted.

7) At line 26 and 27 the Authors use the terms “superior” and “lower”. I would suggest adding a quantification to these terms.

8) At line 57, I would suggest adding literature citations to support the two previous sentences.

9) At line 78 the Authors mention that to their knowledge, no previous study has reported the use of vessel wall MRI in differentiating intracranial aneurysms from the lesions suspected of being aneurysms.

I found a few interesting papers on the topics that may be worth referencing, even if not necessarily at line 78:

DOI: 10.7461/jcen.2015.17.3.180

DOI: 10.1136/neurintsurg-2019-015149

DOI: https://doi.org/10.3174/ajnr.A6080

DOI: https://doi.org/10.3174/ajnr.A4893

10) At line 104 the acronym “ICA” is used by the Authors, however I could not find an explanation of what the acronym stands for in the text of the manuscript.

11) Sentence starting at line 106 “Index lesions with an ophthalmic artery origin are lesions that appeared to be in contact with the ophthalmic arteries on volume-rendered TOF-MRA images.” Could the Authors clarify the purpose of this sentence and the connection to the previous and following sentences?

12) Table 1:

• Please correct the alignment of the words “Index lesions” and “Aneurysms” in the two columns.

• Under the columns referring to “Size without ophthalmic artery origin”, how were the new totals (100% of the 2 columns below) calculated?

• Under the columns referring to “Size without ophthalmic artery origin”, the second column total doesn’t add up to 100.

• Under “Location”, the second column total doesn’t add up to 100.

13) At line 119, I would suggest the word “machines” to be substituted with the word “systems”.

14) From line 126 to 128, please check the units. Why do the matrix dimensions have a unit of mm (as per my experience it should be unitless)? FOV unit should be mm^3, as well as for voxel size.

Same suggestions for line 137 to 140.

15) At lines 148-150, please check the units and dimensions.

16) At line 160 the Authors mention that the neuroradiologists reviewed the images, but it is not specified how many times they reviewed them. I believe adding this information would make clearer how the total of 1356 observations was obtained. I inferred the total from Table 2; however I couldn’t find it throughout the text.

17) At line 281 the Authors use the term “good”. I would suggest choosing a different word, as “good” can be considered as a subjective (non-quantitative) term.

Reviewer #2: The study investigates the additional diagnostic value of 3D Proton Density Imaging in the evaluation of indeterminate vessel wall lesions.

The technique seems interesting in this context, as high resolution PD-vessel images can potentially be achieved at a reasonable acquisition time (at least in 2/3 protocols).

The study, although retrospective, seems to have been carried out thoroughly and certainly includes a large number of cases, which allows a statement about the method.

The study needs a fundamental revision, in particular because important references were not mentioned, which need to be addressed.

1) The study does not mention that recently a very similar study involving PD imaging has been published (first study from Seoul):

Kim S, Chung J, Cha J, et al. Usefulness of high-resolution three-dimensional proton density-weighted turbo spin-echo MRI in distinguishing a junctional dilatation from an intracranial aneurysm of the posterior communicating artery: a pilot study. Journal of NeuroInterventional Surgery 2020;12:315-319

Seon Jin Yoon, Na-Young Shin, Jae Whan Lee, Seung Kon Hu and Keun Young Park. Localization and Treatment of Unruptured Paraclinoid Aneurysms: A Proton Density MRI-based Study. J Cerebrovasc Endovasc Neurosurg. 2015;17(3):180-184

The study submitted here must indicate whether it is a patient cohort independent of the above-mentioned studies (concern about dual publication).

The study seems to go beyond these publications both in terms of the location of the lesions and the number of cases. Please comment on the progress of your study compared to these studies.

2) L3-4: The Background paragraph needs rephrasing

3) The detection and confidence of aneurysm/infundibulum visualization is certainly a function of spatial resolution. This should be adressed in the introduction and discussion sections. TOF MRA benefits from compressed-sense techniques, improving spatial resolution at reasonable scan times. This should be adressed in the discussion, as the presented TOF technique seems not to represent latest state-of-the art technique. Please do include current references on TOF MRA imaging.

4) Is it possible to define a technique-dependent cutoff-value regarding aneurysm size based on the ROC-analysis? I suppose that this is complicated by the different resolution being used/clustered data but would increase the value of the study, as probably smaller aneurysms were detectable with additional PD-sequence compared to TOF alone.

Otherwise, it would be necessary to address the different spatial resolutions being used instead of performing a pooled analysis, the cohort size should allow this to be done.

5) Tables 2-4 need reformatting; e.g. fuse tables 3 and 4

6) L283-284: needs rephrasing, with this sentence the mentioned study is not adequately cited

7) L290-292: please change this sentence and also adress Kim S et al. Journal of NeuroInterventional Surgery 2020;12:315-319

8) L295-298: Please mention the spatial resolution/voxel size, which seems smaller compared to TOF MRA. Hypothesis: does PD MRA benefit from true isotropic acquisition (compared to TOF?) and higher CNR?

9) Please also adress aneurysm location (e.g. skull base, ca) as a relevant factor regarding aneurysm detection in the discussion section. Was detectability of „true“ aneurysms higher in these locations when using additional PD?

10) The ROC-curves are not being presented in the study, would certainly improve its value.

11) Please revise figures 1-3: mention image planes in the figure legend. The illustrations also need a general didactic revision with regard to the recognizability of the pathology.

a. Fig. 1: a,b please zoom in; 1c-h mention image plane in the figure legend

b. Fig. 2: c-j mention image plane in the figure legend

c. Fig. 3: a,b please zoom in; c-j mention image plane in the figure legend

6. PLOS authors have the option to publish the peer review history of their article (what does this mean?). If published, this will include your full peer review and any attached files.

Reviewer #1: No

Reviewer #2: No

---

## [Author Response · Author response to Decision Letter 0]

21 Aug 2020

Journal Requirements 

Answer) As required, we have revised our manuscript based on the PLOS ONE style templates

2. Please clarify in your Data availability statement how other researchers can obtain the same dataset. For PLOS ONE data sharing guidelines, please see https://journals.plos.org/plosone/s/data-availability"

Answer) First, the patients enrolled in our study did not overlap with those included in the previously published articles, which may seem similar to our study; however, our study was conducted independently. Our article was already presented at previous conferences held in 2018 (European congress of Radiology 2018 and Asia-Oceanian Congress of Neuroradiology 2018); therefore, we previously commented that there was no previously published report during writing this manuscript. However, we deleted the comments declaring our study as the first report on the subject and added the corresponding references to our list in the manuscript.

3. Thank you for including your ethics statement:

"This retrospective study was approved by our institutional review board, which waived the requirement for informed consent. ".

i) Please amend your current ethics statement to include the full name of the ethics committee/institutional review board(s) that approved your specific study.

Answer) We changed the ethics statement as follows:

"This retrospective study was approved by Asan Medical Center institutional review board that waived the requirement for informed consent."

Editor’s requirements

1. This manuscript was evaluated with substantial criticism, in particular with regards of omission of important recent references of very similar studies on PD imaging of intracranial aneurysms by Korean groups:

1. Kim S, Chung J, Cha J, et al. Usefulness of high-resolution three-dimensional proton density-weighted turbo spin-echo MRI in distinguishing a junctional dilatation from an intracranial aneurysm of the posterior communicating artery: a pilot study. Journal of NeuroInterventional Surgery 2020;12:315-319

2. Seon Jin Yoon, Na-Young Shin, Jae Whan Lee, Seung Kon Hu and Keun Young Park. Localization and Treatment of Unruptured Paraclinoid Aneurysms: A Proton Density MRI-based Study. J Cerebrovasc Endovasc Neurosurg. 2015;17(3):180-184

There appears to be a serious concern of dual publication of data and it appears unclear whether these are publications from the same group or not. It is not understandable why the authors did not mention these publications from colleagues on the same topic!

So we cannot accept this paper in PLOS ONE if the authors do not make clear that:

1. there is no overlap in analyzed patient cohorts between their study submitted to our journal and the other two studies published in different journal.

2. what the added values and differences in methods, analyzed endpoints/variables and results between their submitted studied and the two previously published studies was!

Answer) Thank you for your important comment. First, the patients enrolled in our study did not overlap with those included in the two aforementioned articles because our study was conducted independently. Our article was already presented at previous conferences held in 2018 (European congress of Radiology 2018 and Asia-Oceanian Congress of Neuroradiology 2018); therefore, we previously commented that there was no previously published report during writing the manuscript. However, we deleted the comments declaring our study as the first report on the subject and added the corresponding references as follows: 

“However, to our knowledge, no previous study has reported the use of vessel wall MRI in differentiating intracranial aneurysms from the lesions suspected of being aneurysms (the index lesion).” in introduction

“There was one report that demonstrated the performance of proton density images for identifying aneurysms of the infundibula. However, Kim et al. discussed that the diagnostic performance of 3D-PD MRI was found to be better than TOF-MRA only in the lesions of the posterior communicating arteries within a small population [21]. Yoon et al. found that PD-MRI can disclose the detailed location (intradural, extradural, or transdural) of paraclinoid internal carotid artery aneurysms with reference to distal dural rings without identifying aneurysms of the infundibula [22].” in Introduction.

“To our knowledge, no previous study has reported the use of vessel wall MRI in the diagnosis or differentiation of aneurysms from infundibula or normal variations.” in discussion

“This study of 299 patients demonstrated the usefulness of 3D-PD MRI compared to that of TOF-MRA in identifying infundibular aneurysms in entire arterial segments of the Circle of Willis, except the paraclinoid internal carotid artery aneurysms. However, Kim et al. presented a pilot study of 82 patients investigating only the lesions of the posterior communicating arteries [21]. Therefore, 3D-PD MRI can be applied widely.” in Discussion

Reviewers' comments: 

Answer) As written above, we have already addressed these concerns. 

Reviewer #1: 

This is an interesting original paper that quantifies with statistical analysis the improvement in diagnostic performance of the TOF-MRA combined with 3D-PD MRI with respect to the TOF-MRA performed alone for the identification of aneurysms and infundibula from the detected lesions in the circle-of-Willis. DSA is used as reference standard to verify the diagnostic outcomes. The methodology and the statistics are sufficiently detailed, and the study is properly placed in the context. The Authors clearly expose the motivation for this study and its use in the context of lifting the patient from the fear and anxiety due to the knowledge of having an aneurysm, when the probability of that lesion actually being an infundibulum is not negligible. Also, the proposed technique is completely non-invasive and would offer additional diagnostic capabilities by using it to complement other non-invasive techniques such as TOF-MRA, which used alone would have lower performance. This is an important aspect to take into consideration, since the gold standard alternative of the DSA carries important risks, as explained by the authors.

Answer) Thank you for your comments. Upon rechecking the data, we recognized that there was a mistake in the previously reported AUC value of TOF-MRA. We had presented 0.814 as the AUC of TOF-MRA but the correct value was 0.837 (0.817-0.857). Hence, we intend to revise the value. We apologize for this mistake.

The image acquisition details used for the different diagnostic techniques are extensively provided by the Authors. However, it would be nice if more information could be added about the use of different imaging parameters with the same technique and in between the techniques, specifying if the changes are compatible with the possibility of a comparison between the techniques in terms of diagnostic capabilities. For example, can a possible change in resolution have affected the results?

Answer) Thank you for your valuable comment. There are many factors that affect the results, such as resolution, iso- or anisometric acquisition, acceleration factors, and so on. Unfortunately, due to the limited scope of the study, we cannot comment regarding the technical issues associated with smaller portions of higher resolution images and the same imaging acquisition in most cases. However, we have started another study on the identification of aneurysms using 3D-PD MRI in view of resolution, isometric acquisitions, and acceleration factors. After analyzing the results of the new, we will be able to provide comments on this subject. I think that your suggestions are very important and insightful.

1) At line 317 in the Discussion section the Authors mention that the time elapsed between the PD MRI and the DSA is variable and in some cases up to 6 months. Would it be possible to add an additional comment specifying how this variation could have significantly or not affected the results?

I believe it would also be important at this point to refresh the memory of the readers about the fact that the TOF-MRA and 3D-PD MRI have been performed simultaneously, thus the main comparison between TOF-MRA alone and TOF-MRA combined with 3D-PD MRI is not affected by this time issue.

Regarding the comparison with DSA instead, which would be affected by the time issue, my understanding is that if DSA was performed afterwards, and only a worsening of the situation can be considered with time, the calculated performance compared to DSA could have only been better than the shown results in the case where the DSA had been performed without waiting a long time.

Answer) Thank you for your valuable comment. In most cases, DSA was performed after MRI. The time intervals between MRI and DSA can contribute to the change in size and morphology of aneurysms, which might affect detectability or diagnostic performances. For example, tiny aneurysms were not identified using MRI but were identified more easily showing growth in size and peculiar morphological changes using DSA. However, we think that the morphological changes were not dynamic especially within the 6-month interval in unruptured intracranial aneurysms. Because, rupture risk of unruptured intracranial aneurysms is known to be 1.1 – 1.3 % in average annual incidence (Natural History of Unruptured Intracranial Aneurysms A Long-term Follow-up Study Stroke 2013; Natural history of unruptured intracranial aneurysms: probability of and risk factors for aneurysm rupture J Neurosurg 2008), which may mean too short interval to occur any change in size and morphology in unruptured aneurysms. However, a change in size and geometry still cannot be excluded. Hence, we wrote “it remains unclear whether the changes in size and morphology did not affect the diagnostic performances.” 

2) At line 7, term “index lesions”. I see the Authors give a definition of the term at line 80 in the Introduction section but, using this term in the abstract without a definition seems confusing to me. I would suggest providing an explanation for this from the beginning.

Answer) Thank you for your comment. We added the explanation in the abstract as follows: “index lesions refer to vascular bulging lesions without vessels arising from the apex, observed using volume-rendered TOF-MRA”.

3) Multiple times throughout the manuscript the Authors use the term “differentiating”. An instance can be found at line 81. The word "differentiating" as it is used in the sentences leads me to think that the Authors want to differentiate between the aneurysm and the index lesion, while they are actually identifying which of the index lesion is an aneurysm or an infundibulum. Therefore, I would rather use the word identifying, by saying "identifying the true unruptured aneurysms among the index lesions".

Answer) Thank you for your comment. We changed “differentiating” to “identifying”.

4) Line 14, “Three neuroradiologists”: after reading this in the abstract, it was confusing for me to find more people mentioned in the Materials and Methods section under Image Analysis.

Answer) Thank you for your comments. Three neuroradiologists participated in identifying true unruptured aneurysms among the index lesions. Other two radiologists participated in identifying index lesions and reviewing DSA as a reference standard. Therefore, we changed the sentence in the Abstract to avoid confusion as follows: 

“Final diagnoses were made by another neuroradiologist and neurointerventionist, in consensus, using DSA as the reference standard.” in Abstract

5) Line 16: Is the parenthesis an explanation of what the Authors mean with aneurysm, differently from the infundibula from which instead vessel originates? As it is positioned in the sentence, it seems instead referring to the definition of index lesions.

Answer) Thank you for your comments. We have explained the definition of index lesions according to your suggestion, as indicated above. Hence, we deleted the definition provided within parentheses. 

“index lesions (vascular bulging lesions without vessels arising from the apex on volume-rendered TOF-MRA)”.

6) End of the line 18: I think there is an “of” that would need to be deleted.

Answer) Thank you for your detailed observation. We have deleted “of” accordingly.

7) At line 26 and 27 the Authors use the terms “superior” and “lower”. I would suggest adding a quantification to these terms.

Answer) Thank you for your suggestions. We added the specific numbers as follows:

“3D-PD MRI with TOF-MRA showed superior diagnostic performance (p=.001; accuracy, 85.5 % versus 95.4 %), superior area under the receiver operating characteristic curve over TOF-MRA (p=.001; 0.814 versus 0.947), and a lower proportion of undetermined lesions than TOF-MRA (p=.001; 25.1 % versus 2.3 %).”

8) At line 57, I would suggest adding literature citations to support the two previous sentences.

Answer) Thank you for your suggestion. We added the corresponding references as follows: 

“With the increase in the demand for health checkups and accessibility of advanced imaging techniques, the detection of incidental aneurysms is increasing [3, 4].”

“However, small infundibular dilatations are sometimes confused with true aneurysms [3, 5].”

3. Sun L-J, Li Y-D, Li M-H, Wang W, Gu B-X. Aneurysm outflow angle at MRA as discriminant for accurate diagnosis and differentiation between small sidewall cerebral aneurysms and infundibula. Journal of neurointerventional surgery. 2016:neurintsurg-2016-012425.

4. Wermer MJ, van Walderveen MA, Garpebring A, van Osch MJ, Versluis MJ. 7Tesla MRA for the differentiation between intracranial aneurysms and infundibula. Magnetic Resonance Imaging. 2017;37:16-20.

5. Yang ZL, Ni QQ, Schoepf UJ, De Cecco CN, Lin H, Duguay TM, et al. Small Intracranial Aneurysms: Diagnostic Accuracy of CT Angiography. Radiology. 2017;285(3):941-52. Epub 2017/06/28. doi: 10.1148/radiol.2017162290. PubMed PMID: 28654338.

9) At line 78 the Authors mention that to their knowledge, no previous study has reported the use of vessel wall MRI in differentiating intracranial aneurysms from the lesions suspected of being aneurysms.

I found a few interesting papers on the topics that may be worth referencing, even if not necessarily at line 78:

DOI: 10.7461/jcen.2015.17.3.180

DOI: 10.1136/neurintsurg-2019-015149

DOI: https://doi.org/10.3174/ajnr.A6080

DOI: https://doi.org/10.3174/ajnr.A4893

Answer) Thank you for your important inputs that help us strengthen our study. First, the patients enrolled in our study do not overlap with those from included in the four aforementioned articles because our study was conducted independently. Our article was already presented at previous conferences held in 2018 (European congress of Radiology 2018 and Asia-Oceanian Congress of Neuroradiology 2018). Hence, we had commented that there was no previously published report upon writing the manuscript. However, because of the delayed presentation of our results, we have deleted the comments declaring no previous reports on the subject. There was one report (Surveillance of Unruptured Intracranial Saccular Aneurysms Using Noncontrast 3D-Black-Blood MRI: Comparison of 3D-TOF and Contrast-Enhanced MRA with 3D-DSA) and one review (Intracranial Vessel Wall MRI: Principles and Expert Consensus Recommendations of the American Society of Neuroradiology) about aneurysms using T1W with black-blood technique. However, as you know, the purposes of these studies were different from those of our study. Our study aimed to identify aneurysms using 3D-PD MRI and not to predict the rupture risk or measure the aneurysmal profiles. Furthermore, in detecting or identifying aneurysms, PD MRI is extremely superior to T1W with or without black-blood techniques until now. I think that T1W with or without black-blood techniques is useless in detecting aneurysms. We accordingly added references on PD-MRI and modified our manuscript as follows: 

“However, to our knowledge, no previous study has reported the use of vessel wall MRI in differentiating intracranial aneurysms from the lesions suspected of being aneurysms (the index lesion).” in introduction

“There was one previous report that demonstrated the performance of proton density images in identifying aneurysms of the infundibula. However, Kim et al. presented that diagnostic performance of 3D-PD MRI was found to be better than TOF-MRA only for lesions of the posterior communicating arteries within a small population [21]. Yoon et al. found that PD-MRI can disclose the detailed location (intradural, extradural, or transdural) of paraclinoid internal carotid artery aneurysms with reference to distal dural rings without identifying aneurysms of the infundibula [22].” in Introduction.

“To our knowledge, no previous study has reported the use of vessel wall MRI in the diagnosis or differentiation of aneurysms from infundibula or normal variations.” in discussion

“This study with 299 patients demonstrated the usefulness of 3D-PD MRI compared to that of TOF-MRA in identifying infundibular aneurysms in entire arterial segments of the Circle of Willis, except the paraclinoid internal carotid artery aneurysms. However, Kim et al. presented a pilot study with 82 patients investigating only the lesions of the posterior communicating arteries [21]. Therefore, 3D-PD MRI can be applied widely.” in Discussion

10) At line 104 the acronym “ICA” is used by the Authors, however I could not find an explanation of what the acronym stands for in the text of the manuscript.

Answer) Thank you for your detailed observation and comment. We added the explanation of the first acronym as follows:

“Typical paraclinoid internal carotid artery (ICA) aneurysms”

11) Sentence starting at line 106 “Index lesions with an ophthalmic artery origin are lesions that appeared to be in contact with the ophthalmic arteries on volume-rendered TOF-MRA images.” Could the Authors clarify the purpose of this sentence and the connection to the previous and following sentences?

Answer) Thank you for your comment. In our study, paraclinoid ICA aneurysms were excluded due to the extremely small number of infundibula in the area. However, some locations could sometimes cause confusion in determining paraclinoid and ophthalmic areas. Hence, we defined ophthalmic aneurysms as we mentioned in the manuscript to differentiate it from paraclinoid ICA aneurysms. 

12) Table 1:

• Please correct the alignment of the words “Index lesions” and “Aneurysms” in the two columns.

• Under the columns referring to “Size without ophthalmic artery origin”, how were the new totals (100% of the 2 columns below) calculated?

• Under the columns referring to “Size without ophthalmic artery origin”, the second column total doesn’t add up to 100.

Answer) Thank you for your observations and comments. There was a mistake in rounding off to the nearest hundredths. We intend to delete the column on size without ophthalmic artery origin because we did not analyze the lesions with or without ophthalmic artery origin. 

• Under “Location”, the second column total doesn’t add up to 100.

Answer) Thank you for your comment. There was a mistake in rounding off to the nearest hundredths. We have changed the value from 1.2 to 1.15.

13) At line 119, I would suggest the word “machines” to be substituted with the word “systems”.

Answer) Thank you for your comment. As suggested, we have changed “machines” to “systems.”

14) From line 126 to 128, please check the units. Why do the matrix dimensions have a unit of mm (as per my experience it should be unitless)? FOV unit should be mm^3, as well as for voxel size.

Same suggestions for line 137 to 140.

Answer) Thank you for your suggestions. We apologize for the mistake and have accordingly removed “mm” in the matrix dimensions and changed the units to “mm3” in FOV.

15) At lines 148-150, please check the units and dimensions.

Answer) Thank you for your comment. We apologize for the mistake. We changed “mm” to “mm3” in pixel size.

16) At line 160 the Authors mention that the neuroradiologists reviewed the images, but it is not specified how many times they reviewed them. I believe adding this information would make clearer how the total of 1356 observations was obtained. I inferred the total from Table 2; however I couldn’t find it throughout the text.

Answer) Thank you for your comments. There were 299 patients and a total of 452 (173+276+3) lesions observed in the patients, which is presented in Table 1. The 1356 observations from Table 2 resulted from triplicates of 452 observations because our reviewers comprised three neuroradiologists, which has also been mentioned in the manuscript.

17) At line 281 the Authors use the term “good”. I would suggest choosing a different word, as “good” can be considered as a subjective (non-quantitative) term.

Answer) Thank you for your comment, we have changed the sentence as follows:

“3D-PD MRI can play a role as a supplementary imaging modality for the diagnosis of aneurysms.”

Reviewer #2: The study investigates the additional diagnostic value of 3D Proton Density Imaging in the evaluation of indeterminate vessel wall lesions.

The technique seems interesting in this context, as high resolution PD-vessel images can potentially be achieved at a reasonable acquisition time (at least in 2/3 protocols).

The study, although retrospective, seems to have been carried out thoroughly and certainly includes a large number of cases, which allows a statement about the method.

The study needs a fundamental revision, in particular because important references were not mentioned, which need to be addressed.

1) The study does not mention that recently a very similar study involving PD imaging has been published (first study from Seoul):

Kim S, Chung J, Cha J, et al. Usefulness of high-resolution three-dimensional proton density-weighted turbo spin-echo MRI in distinguishing a junctional dilatation from an intracranial aneurysm of the posterior communicating artery: a pilot study. Journal of NeuroInterventional Surgery 2020;12:315-319

Seon Jin Yoon, Na-Young Shin, Jae Whan Lee, Seung Kon Hu and Keun Young Park. Localization and Treatment of Unruptured Paraclinoid Aneurysms: A Proton Density MRI-based Study. J Cerebrovasc Endovasc Neurosurg. 2015;17(3):180-184

The study submitted here must indicate whether it is a patient cohort independent of the above-mentioned studies (concern about dual publication).

The study seems to go beyond these publications both in terms of the location of the lesions and the number of cases. Please comment on the progress of your study compared to these studies.

Answer) Thank you for these important comments. First, the patients enrolled in our study did not overlap with those included in the two aforementioned articles because our study was conducted independently. Our article was already presented at the previous conferences held in 2018 (European congress of Radiology 2018 and Asia-Oceanian Congress of Neuroradiology 2018); therefore, we previously commented that there was no previously published report during writing the manuscript. However, we deleted the comments declaring our study as the first report on the subject and added corresponding references as follows: 

“However, to our knowledge, no previous study has reported the use of vessel wall MRI in differentiating intracranial aneurysms from the lesions suspected of being aneurysms (the index lesion).” in introduction

“There was one previous report that demonstrated the performance of proton density images for identifying aneurysms of the infundibula. However, Kim et al. discussed that the diagnostic performance of 3D-PD MRI was found to be better than TOF-MRA only in the lesions of the posterior communicating arteries within a small population [21]. Yoon et al. found that PD-MRI can disclose the detailed location (intradural, extradural, or transdural) of paraclinoid internal carotid artery aneurysms with reference to distal dural rings without identifying aneurysms from the infundibula [22].” in Introduction.

“To our knowledge, no previous study has reported the use of vessel wall MRI in the diagnosis or differentiation of aneurysms from infundibula or normal variations.” in discussion

“This study of 299 patients demonstrated the usefulness of 3D-PD MRI compared to that of TOF-MRA in identifying infundibular aneurysms in entire arterial segments of the Circle of Willis, except in paraclinoid internal carotid artery aneurysms. However, Kim et al. presented a pilot study of 82 patients investigating only the lesions of the posterior communicating arteries [21]. Therefore, 3D-PD MRI can be applied widely.” in Discussion

2) L3-4: The Background paragraph needs rephrasing

Answer) Thank you for your comments. We changed the sentences as follows:

“Advanced imaging methods can enhance the identification of aneurysms of the infundibula, which can reduce unnecessary follow-ups or further work-up, fear, and anxiety in patients.”

3) The detection and confidence of aneurysm/infundibulum visualization is certainly a function of spatial resolution. This should be adressed in the introduction and discussion sections. TOF MRA benefits from compressed-sense techniques, improving spatial resolution at reasonable scan times. This should be adressed in the discussion, as the presented TOF technique seems not to represent latest state-of-the art technique. Please do include current references on TOF MRA imaging.

Answer) Thank you for your comments. There are many factors that may affect the results, such as the resolution, iso- or anisometric acquisition, acceleration factors, and so on. The compressed sensing is also a good application for high resolution MRI and TOF-MRA. In our experience (High-Resolution Magnetic Resonance Imaging Using Compressed Sensing for Intracranial and Extracranial Arteries: Comparison with Conventional Parallel Imaging), a spatial resolution with greater than or equal to 0.2x0.2x0.2 mm3 seems difficult to apply in clinical practice with clinical 3T systems in spite of compressed sensing. As you know, recently introduced TOF-MRA sequences with compressed sensing show similar resolution as the conventional TOF-MRA, and most reports focus on the reduction of scan times. Higher resolution to conventional TOF-MRA may mean around 0.1~0.2 mm3 in acquisition voxel size not reconstructed voxel size; however, the associated decreased SNR could be problematic. Thus, it may be not easy to achieve both a higher resolution and shorter scan time simultaneously. As a result, further studies should be conducted in this regard. 

“Recently introduced compressed sensing techniques may contribute to the enhancement of spatial resolution with preserved scan times and the reduction of scan time with preserved image quality in both TOF-MRA [30, 31] and vessel wall MRI [32]. TOF-MRA with a higher resolution involving a reasonable scan time may enhance the identification of aneurysms of the infundibula even though a smaller spatial resolution can have a demerit of lower signal-to-noise ratio [33]. However, recent research on TOF-MRA has focused on the reduction of scan times, and not on the enhancement of spatial resolution [30, 31]. Moreover, even 7T system shows a resolution of 0.23- 0.31 mm3 [4, 34]. Therefore, further studies are necessary.” in the Discussion

“Finally, TOF-MRA was acquired with anisotropic voxels and variable parameters, and was observed to have a slightly larger spatial resolution in the z-axis. The heterogeneity of detailed parameters and differences in spatial resolutions may have affected the results of this study. Therefore, further studies involving equal imaging conditions are warranted” in the limitations

4) Is it possible to define a technique-dependent cutoff-value regarding aneurysm size based on the ROC-analysis? I suppose that this is complicated by the different resolution being used/clustered data but would increase the value of the study, as probably smaller aneurysms were detectable with additional PD-sequence compared to TOF alone.

Otherwise, it would be necessary to address the different spatial resolutions being used instead of performing a pooled analysis, the cohort size should allow this to be done.

Answer) Thank you for your comment. We analyzed our results with a 3-mm reference but no significant differences were observed. As you observed, our data was heterogeneous in location, size, and imaging parameters, which made it difficult to analyze, compare, and stratify the results according to each issue. Thus, we opted to study imaging conditions that were similar between TOF-MRA and PD-MRI, which made it possible to analyze results in terms of location, size, etc. In addition, we added the following statement:

“Finally, TOF-MRA was acquired with anisotropic voxels and variable parameters, and was observed to have a slightly larger spatial resolution in the z-axis. The heterogeneity of detailed parameters and differences in spatial resolutions may have affected the results of this study. Therefore, further studies involving equal imaging conditions are warranted” as limitations

5) Tables 2-4 need reformatting; e.g. fuse tables 3 and 4

Answer) Thank you for your comments. We fused tables 3 and 4 as follows:

Table 3. Comparison of diagnostic performances and proportions of undetermined lesions

 AUC P value

TOF-MRA 0.837

† (0.817 – 0.857) .001

3D-PD MRI with TOF-MRA 0.947

† (0.934 - 0.958) 

 Proportion P value

TOF-MRA 25.1%

†† (341/1356) .001

3D-PD MRI with TOF-MRA 2.3 %

†† (31/1356) 

* TOF-MRA = time-of-flight magnetic resonance angiography, 3D-PD MRI = three-dimensional high-resolution proton density-weighted magnetic resonance imaging, AUC = area under the receiver operating characteristic curve.

† Data in parentheses indicate 95% confidence intervals.

†† Data in parentheses indicate numbers of lesions.

††† P-values were calculated from comparisons between 3D-PD MRI and TOF-MRA.

6) L283-284: needs rephrasing, with this sentence the mentioned study is not adequately cited

Answer) Thank you for your comments. We have changed the references as follows:

25. Hwang SB, Kwak HS, Han YM, Chung GH. Detection of intracranial aneurysms using three-dimensional multidetector-row CT angiography: is bone subtraction necessary? Eur J Radiol. 2011;79(2):e18-23. doi: 10.1016/j.ejrad.2010.01.004. PubMed PMID: 20144517.

3. Sun L-J, Li Y-D, Li M-H, Wang W, Gu B-X. Aneurysm outflow angle at MRA as discriminant for accurate diagnosis and differentiation between small sidewall cerebral aneurysms and infundibula. Journal of neurointerventional surgery. 2016:neurintsurg-2016-012425.

4. Wermer MJ, van Walderveen MA, Garpebring A, van Osch MJ, Versluis MJ. 7Tesla MRA for the differentiation between intracranial aneurysms and infundibula. Magnetic Resonance Imaging. 2017;37:16-20.

7) L290-292: please change this sentence and also adress Kim S et al. Journal of NeuroInterventional Surgery 2020;12:315-319

Answer) Thank you for your comments. We removed the previous sentence and added new sentences with the following references:

“To our knowledge, no previous study has reported the use of vessel wall MRI in the diagnosis or differentiation of aneurysms from infundibula or normal variations.” in discussion

“This study of 299 patients demonstrated the usefulness of 3D-PD MRI compared to that of TOF-MRA in identifying infundibular aneurysms in the entire arterial segments of the Circle-of-Willis except the paraclinoid internal carotid artery aneurysms. However, Kim et al. presented a pilot study of 82 patients investigating only the lesions of the posterior communicating arteries [21]. Therefore, 3D-PD MRI can be applied widely.” in the discussion

8) L295-298: Please mention the spatial resolution/voxel size, which seems smaller compared to TOF MRA. Hypothesis: does PD MRA benefit from true isotropic acquisition (compared to TOF?) and higher CNR?

Answer) Thank you for your comment. The spatial resolution is important in identifying aneurysms. However, in our retrospective study, the resolutions were not the same between PD and TOF-MRA and among TOF-MRA sequences, as you have observed. Thus, we included this issue in the limitations as follows:

“Finally, TOF-MRA was acquired with anisotropic voxels with variable parameters and was found to have a slightly larger spatial resolution in the Z-axis. The heterogeneity of detailed parameters and differences in spatial resolutions may have affected the results of this study. Therefore, further study involving equal imaging conditions is warranted” in the limitations

9) Please also address aneurysm location (e.g. skull base, ca) as a relevant factor regarding aneurysm detection in the discussion section. Was detectability of „true“ aneurysms higher in these locations when using additional PD?

Answer) Thank you for your valuable comment. We did not analyze our results in terms of the location. Thus, we compared the diagnostic performances according to the location (Acom, oph, Pcom, AchA; we did not analyze these because of insufficient data in BA, SCA, ICA, A1) based on the AUC. A difference of AUC in the ophthalmic artery origin aneurysms was the lowest, but there were statistically significant differences across Acom, oph, Pcom, AchA. We have mentioned this issue in the discussion as follows:

“3D-PD MRI combined with TOF-MRA showed a larger AUC compared to that of TOF-MRA alone in the ophthalmic artery origin, posterior communicating artery, anterior communicating artery, and anterior choroidal artery origin locations (p=.033 to .001).” in the results. 

10) The ROC-curves are not being presented in the study, would certainly improve its value.

Answer) Thank you for your suggestion. We have added the ROC curves as seen in Figure 4. In addition, during rechecking of the presented data, we noted a mistake in the AUC value of TOF-MRA. We had presented 0.814 as the AUC of TOF-MRA but the correct value was 0.837 (0.817-0.857). Thus, we intend to revise the value. We sincerely apologize for this mistake.

Fig 4. 3D-PD MRI combined with TOF-MRA (AUC= 0.837) showed higher diagnostic performance and AUC than TOF-MRA alone (AUC= 0.947) (p=.001).

11) Please revise figures 1-3: mention image planes in the figure legend. The illustrations also need a general didactic revision with regard to the recognizability of the pathology.

a. Fig. 1: a,b please zoom in; 1c-h mention image plane in the figure legend

b. Fig. 2: c-j mention image plane in the figure legend

c. Fig. 3: a,b please zoom in; c-j mention image plane in the figure legend

 Answer) Thank you for your comments. We modified figure 1 and 3 using zoomed images and figure legends as follows:

Fig 1. 

“A 60-year-old female was referred from an outside hospital for incidentally found aneurysms. TOF-MRA (a) and TOF-MRA source images (c-e) showed an index lesion around the left anterior choroidal artery origin; however, the relationship between the aneurysm and anterior choroidal artery was unclear on coronal multi-planar reconstruction (MPR) images (arrows). 3D-PD MRI clearly depicted the aneurysm and anterior choroidal artery origin on coronal MPR images (arrows) (f-h). 3D rotational angiography (b) also showed a clear relationship between the aneurysm and the anterior choroidal artery origin (arrow).” 

Fig 2. 

“A 66-year-old female underwent 3D-PD MRI for evaluation of a right middle cerebral artery stenosis detected during work up for left side weakness. TOF-MRA (a) showed an index lesion around the left distal ICA. However, it was difficult to diagnose the aneurysm due to its tiny size and nearby bone structures (arrows) (c–f) even on TOF-MRA source images with oblique coronal multi-planar reconstruction (MPR) (d-f). 3D-PD MRI depicted a definite aneurysm on oblique coronal MPR images (arrows) (g-j). DSA also showed a definite aneurysm (arrow) (b).”

Fig 3. 

“A 57-year-old female underwent 3D-PD MRI for evaluation of a left paraclinoid internal carotid artery aneurysm. TOF-MRA (a) showed an index lesion around anterior choroidal artery origin of left distal ICA. However, it was difficult to diagnose the aneurysm (arrows) even on TOF-MRA source images with sagittal multi-planar reconstruction (MPR) (c-f). The relationship between anterior choroidal artery and the index lesion was not seen clearly on sagittal MPR images of TOF-MRA (e, f). 3D-PD MRI depicted a definite aneurysm with bi-lobed shape (arrows) (g, h) and clear relationship between anterior choroidal artery and the index lesion on sagittal MPR images (i, j). 3D rotational angiography also showed a definite aneurysm with bi-lobed shape (arrow) (b).”

---

## [Decision Letter · Decision Letter 1]

18 Nov 2020

Added diagnostic values of three-dimensional high-resolution proton density-weighted magnetic resonance imaging for unruptured intracranial aneurysms in the circle-of-Willis: Comparison with time-of-flight magnetic resonance angiography.

PONE-D-20-13596R1

Dear Dr. Jung,

We’re pleased to inform you that your manuscript has been judged scientifically suitable for publication and will be formally accepted for publication once it meets all outstanding technical requirements.

Kind regards,

Nicola Toschi

Academic Editor

PLOS ONE

Additional Editor Comments (optional):

Reviewers' comments:

Reviewer's Responses to Questions

**Comments to the Author**

1. If the authors have adequately addressed your comments raised in a previous round of review and you feel that this manuscript is now acceptable for publication, you may indicate that here to bypass the “Comments to the Author” section, enter your conflict of interest statement in the “Confidential to Editor” section, and submit your "Accept" recommendation.

Reviewer #1: All comments have been addressed

Reviewer #2: All comments have been addressed

2. Is the manuscript technically sound, and do the data support the conclusions?

Reviewer #1: Yes

Reviewer #2: Yes

3. Has the statistical analysis been performed appropriately and rigorously? 

Reviewer #1: Yes

Reviewer #2: Yes

4. Have the authors made all data underlying the findings in their manuscript fully available?

Reviewer #1: Yes

Reviewer #2: No

5. Is the manuscript presented in an intelligible fashion and written in standard English?

Reviewer #1: Yes

Reviewer #2: Yes

6. Review Comments to the Author

Reviewer #1: no further comments.............................................................................................................

Reviewer #2: The authors adressed all my comments and recommendations. I only recommend correct reading by a native speaker.

7. PLOS authors have the option to publish the peer review history of their article (what does this mean?). If published, this will include your full peer review and any attached files.

Reviewer #1: No

Reviewer #2: No

---

## [Editor Report · Acceptance letter]

20 Nov 2020

PONE-D-20-13596R1 

Added diagnostic values of three-dimensional high-resolution proton density-weighted magnetic resonance imaging for unruptured intracranial aneurysms in the circle-of-Willis: Comparison with time-of-flight magnetic resonance angiography 

Dear Dr. Jung:

I'm pleased to inform you that your manuscript has been deemed suitable for publication in PLOS ONE. Congratulations! Your manuscript is now with our production department. 

Kind regards, 

on behalf of

Dr. Nicola Toschi 

Academic Editor

PLOS ONE